# DeCAL Tokenwise Compression

## Abstract

This paper introduces DeCAL, a new method for tokenwise compression. De-CAL uses a native encoder-decoder language model pretrained with denoising to learn to produce high-quality compressed representations from the encoder. De-CAL applies small modifications to the encoder, with the emphasis on maximizing compression quality, even at the expense of compute. We show that DeCAL at 2x compression can match uncompressed, with usually only a minor dropoff in metrics up to 8x compression, among question-answering, summarization, and multi-vector retrieval tasks. DeCAL distinguishes itself from prior works by supporting larger chunks and significantly more compressed output per chunk.

## 1 Introduction

Transformers (Vaswani et al., 2017) have proven very effective at a broad range of natural language tasks, but inference costs increase dramatically as sequence lengths increase. Efforts to address the cost often focus on speeding up the attention block (Tay et al., 2022), the main bottleneck, usually by sparsifying compute, but retaining all tokens since it is unknown which subsequent attention computations might attend to them. Pruning input tokens (Tang et al., 2024) is another strategy, but that leads to forgetting, which may sacrifice performance on certain tasks.

Alternatively, we can compress the input sequence by merging multiple token representations, potentially retaining more information. Token sequences should be fairly compressible given that the subwords they are built from are nonuniform in their information content and follow predictable patterns. Individually, subwords do not contain much information, but in sequence together they do; however, that information does not require their original space. Hence, the opportunity for compression is clear.

Approaches for tokenwise compression rely on variants of attention pooling (as in Funnel Transformer (Dai et al., 2020)) and convolution (as in CANINE (Clark et al., 2022)), primarily to deliver substantial training and inference speedups. The speedups do come at a cost to model performance, as we have observed in practice with the Funnel Transformer.

More recent work such as AutoCompressor (Chevalier et al., 2023), COCOM (Rau et al., 2025), and ICAE (Ge et al., 2024) share the goal of LLM context compression. They all derive compressive encoders from decoder-only LLMs in order to fit more information within the context budget of the LLM, as well as save inference time when the context chunks can be precomputed. We will refer to this class of models as **retrofitted** context compressors.

We present DeCAL, a novel method for tokenwise compression. DeCAL employs an encoder structure in which we prepend an input-derived latent sequence to the encoder input. This shorter latent sequence successively attends to the evolving input sequence, layer by layer, but only the latent is output. This structure forms the basis of the name DeCAL, short for Deep Cross-Attended Latents. DeCAL focuses on maximizing compressed representation quality, without making tradeoffs for immediate inference speedup, leaving compute optimization for later.

In contrast to retrofitted compressors, DeCAL is built **natively** on an encoder-decoder platform, T5 (Raffel et al., 2020), to perform context compression from the ground up. This choice facilitates exploring deeper into the design space, including training a customized compressive encoder alongside the decoder from the start of pretraining.

Our main contributions are as follows:

- We train DeCAL (as an encoder-decoder language model) from scratch with compression on a denoising language modeling task. We show that this is superior to the more typical approach of autoencoding (mixed with text continuation) after standard pretraining.

- We initialize DeCAL's latent sequence with pooled input tokens combined with a learnable embedding vector, and show that this outperforms using the learnable vector alone.

- A simple alternative to DeCAL's prepending $m$ latent tokens to the input sequence is to instead perform attention pooling on the output layer down to $m$ tokens. Though the latter is faster, we establish that it delivers inferior compression.

- We fine-tune DeCAL on context-intensive tasks, namely summarization and question answering, and compare different levels of compression, with only minor metrics dropoff at 8x compression. We also construct a proxy for typical LLM context compression approaches to compare against and determine that the proxy falls short of DeCAL performance.

- We illustrate the versatility of DeCAL by additionally applying it to multi-vector retrieval tasks.

- Lastly, we examine several limitations of the retrofitted context compressors that DeCAL is able to address. Specifically, DeCAL can support a larger input chunk size, can output many more compressed tokens, and can serve better for summarization tasks.

## 2 RELATED WORK

There are a number of studies that involve compressing token sequences, though most do this internally for efficiency, not as an explicit usable output sequence.

**Compressive Encoder-only.** The Funnel Transformer (Dai et al., 2020) applies multiple steps of attention-pooling compression, followed by a single step of upsampling back to full sequence length (plus residual from the last full length layer). The upsampled layers are used for masked language pretraining, then discarded. In contrast, DeCAL does not compress the input stepwise in-line to the output, but rather cumulatively, alongside the input. DeCAL uses an autoregressive decoder to interpret the encoder's compressed output and so more naturally handles more complex language modeling such as variable-length corrupted token spans.

Perceiver (Jaegle et al., 2021) uses a short latent sequence that repeatedly cross-attends to the original input sequence, thus requiring much less compute. DeCAL instead combines the latent with the original input and both sequences evolve layer by layer, requiring net additional compute (trading off for higher compression quality).

**Internal compression.** Some studies use compression internally, but not in any externally usable form. CANINE (Clark et al., 2022) applies convolutions for both downsampling and upsampling (but no residual connections), with the goal of handling character-level inputs. Byte Latent Transformer (Pagnoni et al., 2024) groups input bytes into variable length patches which undergo attention pooling into one global token per patch before being upsampled with reversed attention pooling (the last full-length layer output serves as the attention queries). Both approaches apply the bulk of their compute in their compressed inner layers.

**History compression.** Additional studies compress past history of decoders to cope with very long contexts. The Compressive Transformer (Rae et al., 2019) propagates past token history through recurrent encoding of token blocks, and then extends its memory capacity by compressing (via convolution) all but the last block. The Recurrent Memory Transformer (Bulatov et al., 2024) also performs recurrent encoding of token blocks, but carries over only a small fixed number of tokens per block, for very high compression to achieve 1M+ token contexts, but is very task-specific.

**Pruning.** A less common technique to shorten sequences is pruning tokens, such as PoWER-BERT (Goyal et al., 2020), which drops a few tokens every layer, knowing that at the output, only the CLS token is used. However, this is only added after pretrain and fine-tune, then an auxiliary model is used to identify how to drop tokens, resulting in a task-specific inference speedup. NUGGET (Qin & Van Durme, 2023) has an encoder-decoder format, but adds small FFNs on the encoder outputs to perform top-K selection. NUGGET is trained as an autoencoder, but is limited to 128 token blocks.

**Soft prompts.** Soft prompts (Li & Liang, 2021) involve adding a learnable prefix to a frozen model's input, using a different prefix per task as an alternative to separate fine-tuning per task. This looks superficially like DeCAL in its learnable prefix, but is functionally orthogonal; DeCAL's encoder outputs only the latent sequence, which is a function of the input sequence, unlike soft prompts.

**Context compression.** AutoCompressors (Chevalier et al., 2023) build on the Recurrent Memory Transformer (Bulatov et al., 2024) by keeping prior blocks' summary vectors and successively concatenating them to finally obtain an effective compressed form that is used as a soft prompt. The In-context Autoencoder (ICAE) (Ge et al., 2024) works similarly, but compresses blocks in parallel instead of recursively, so blocks do not see their neighboring context. ICAE's compression training consists of autoencoding with text continuation, in contrast to Autocompressor's next token prediction. COCOM (Rau et al., 2025) develops further by unfreezing the decoder, and achieves a better compression ratio than ICAE, but encodes smaller chunks.

All of these works share the common approach of appending special context tokens to accumulate the compressed output representation, which is similar to DeCAL, though DeCAL initializes the latent tokens differently. All adapt from decoder-only LLMs, and do not apply denoising during their compression training, unlike DeCAL.

It is worth observing the commonality that the underlying compression operation in all of the above techniques is most often cross-attention from a small sequence to a larger one and, in some cases, convolution.

## 3 METHODS

The essential elements of DeCAL are:

- An encoder-decoder model configuration
- A compressive encoder structure with latent sequence prepended to the input
- Pooling of input tokens to seed the initial latent sequence state
- A denoising language modeling pretrain task

In an encoder-decoder language model, as in T5 (Raffel et al., 2020), the encoder provides the input context, while the decoder makes next-token predictions. This provides two very useful properties: 1) Unlike encoder-only masked-LM (Devlin et al., 2019), there are no indicators of how many tokens a compressed input token originated from, nor any residuals of the uncompressed input; the decoder is cleanly decoupled and must discern the contents of each compressed token. This puts more weight on the encoder to do a better job of compression and on the decoder to validate that. 2) Training examples have separate sequences for the encoder and decoder, allowing us to tune the amount of work each must do.

We find that the standard span corruption language modeling task in T5 (Raffel et al., 2020) works very well. For the encoder input, it removes spans of tokens, replacing them with sentinel tokens. The decoder target sequence is the complement, with just the tokens that had been removed, and sentinel tokens between them representing the token spans provided by the encoder. So despite the input to the decoder being compressed, it must still identify the sentinel tokens and the preceding and succeeding tokens, then denoise to output the correct span. In principle, we can expect this denoising task to outperform autoencoding. The reason is that during autoencoding, the decoder's autoregressive output contains all the uncompressed tokens so far. This allows the decoder to use past uncompressed tokens in addition to the compressed input to predict the next uncompressed token. This crutch weakens the task, while this uncompressed past state will not be available for the ultimate downstream tasks. In contrast, the T5 denoising task has disjoint encoder input and decoder target tokens, ensuring that the encoder's output is never overshadowed.

DeCAL is constructed as follows, using a standard T5 encoder-decoder:

- We take $\boldsymbol{x} = [x_1, ..., x_n]$ as the input embeddings to be compressed.
- To create the initial latent sequence $\boldsymbol{l} = [l_1, ..., l_m]$ where $m < n$ (giving us a compression ratio $C = n/m$),

- We start with a learnable vector $\boldsymbol{v}$ repeated $m$ times.
- We then add the input embeddings, pooled from length $n$ to $m$ by pooling function $P$. In this work, for $P$, we use a strided mean pool with window size $C$ (same as Funnel Transformer's (Dai et al., 2020) attention query in its attention pooling).
- Lastly, we apply a layer norm.
- Thus $\boldsymbol{l} = LayerNorm(\boldsymbol{v}_{\times m} + P(\boldsymbol{x}))$

- The input to the encoder is $\boldsymbol{l} \oplus \boldsymbol{x}$, thus longer than the original input, as shown in Figure 1.
- The encoder is internally unmodified, so there are negligible additional parameters. Note that both the $\boldsymbol{l}$ and $\boldsymbol{x}$ subsequences are able to attend to each other as their representations are updated in each layer.
- We assign the position index of tokens in $\boldsymbol{l}$ to be the integer mean of the position indices of the corresponding tokens in $\boldsymbol{x}$. This is used for T5 relative position calculations.
- The encoder output is exclusively the $m$ output embeddings corresponding to $\boldsymbol{l}$, as shown in Figure 1.
- The decoder is unchanged.

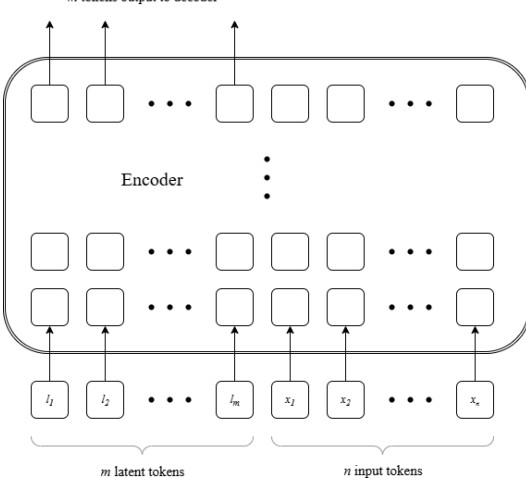

Figure 1: Diagram of DeCAL encoder input and output for compressing $n$ input tokens to $m$ output tokens.

In this scheme, we are logically assigning each token in $\boldsymbol{l}$ to a span of tokens in $\boldsymbol{x}$, forming the correspondence of which $\boldsymbol{x}$ tokens we expect to be compressed into which latent token. We initialize $\boldsymbol{l}$ on this basis, but note we do nothing further with this correspondence, such as attention masking, etc. In our experiments, we only use a constant C, but the design can easily accommodate variable length mappings, such as used in BLT (Pagnoni et al., 2024). $\boldsymbol{l}$ starts only from initial token embeddings, but as the $\boldsymbol{x}$ subsequence is updated after each transformer layer, the $\boldsymbol{l}$ subsequence can attend to $\boldsymbol{x}$ as well as itself, building up a compressed contextualized representation. Likewise $\boldsymbol{x}$ is able to attend to $\boldsymbol{l}$, though we haven't tested its utility. We observe that this approach increases compute cost due to the increase in encoder input length; interestingly, this means compressing more (shorter $\boldsymbol{l}$) is lower compute than compressing less (longer $\boldsymbol{l}$), which has to be taken into account when trading off compute against compression quality.

The encoder maintains constant compression C, scaling $m$ as necessary if $n$ changes; e.g. if after pretraining at 4x and 1024 token inputs for example, we then fine-tune the model on a task with 8192 token inputs, the encoder will output 2048 tokens. Note that we only use a single learnable vector to identify the latent tokens to the encoder, not one vector per latent token; this enables DeCAL to readily generalize to different sequence lengths during finetuning. Our handling of position ids for latent tokens plus token pooling suffices to differentiate individual latent tokens to the encoder.

It would seem logical that performing compression alongside a normal transformer stack would require differing parameters; we tried having separate parameters for the transformer blocks handling

the latent tokens, however, this did not produce better results, so further tweaking may be required to continue down that path.

# 4 EXPERIMENTS

## 4.1 SETUP

All our experiments used the T5.1.1[1] Large encoder-decoder, pretrained from scratch on 128M examples from C4 with a sequence length of 1024, and the default span corruption configuration of 15% corruption, which means 85% of the training tokens are fed to the encoder and 15% to the decoder. We used 10k steps of linear warmup and used a 4x larger relative position `num_buckets` and `max_distance` than T5's default, since that worked best. The DeCAL models were trained identically, with the model structure as described under Methods and compression ratios varying from 2x to 8x.

## 4.2 ENCODER-DECODER EXPERIMENTS

### 4.2.1 DATASETS

Since this work is focused on compressing encoder inputs, we focused on summarization and question-answering tasks, which are naturally context-intensive, to evaluate compression effectiveness. For summarization, we used the arXiv (Cohan et al., 2018) and PubMed (Cohan et al., 2018) datasets, for which the document is the encoder input and the abstract is the target summary. We also included CNN / DailyMail (Nallapati et al., 2016), where the news article is the encoder input and the article's summary bullets are the target summary, as well as MediaSum (Zhu et al., 2021), where interview transcripts are the encoder input and their topic and overviews served as the target summary. For question-answering, we used HotpotQA (Yang et al., 2018) and TriviaQA (Joshi et al., 2017), for which the question and article context are given to the encoder, and the (first) answer is the target.

We fine-tuned on 6M examples, except arXiv, for which we trained on 12M examples, since that task took longer to converge. We limited context to 4k tokens for all tasks due to memory limits and did not apply any chunking. Note that this means all experiments had the same amount of original context information; e.g. our 8x compressed encoders had 4k input tokens and output 512 to the decoder. Thus the metrics reflect only the amount of information loss due to compression.

### 4.2.2 LANGUAGE MODELING

We tested DeCAL with compression ratios of 2x, 4x, and 8x. Since these ratios get quite aggressive, it can be instructive to see how these impact encoder-decoder pretraining. As shown in Table 1, the DeCAL target token prediction accuracy at 2x is able to match the (uncompressed) vanilla T5 Large baseline, while 4x is sufficiently close that further training may close much of the gap. At 8x, accuracy takes another comparable step down. Training at very high compression ratios of 16x and higher proved unstable during pretraining.

### 4.2.3 COMPARISONS

Having a basis of comparison with prior work is useful in placing new work in context. The methods that are most similar to DeCAL are the retrofitted LLM context compressors, since they share the mechanism of adding extra tokens to aggregate information. Specifically, AutoCompressor (Chevalier et al., 2023), COCOM (Rau et al., 2025), and ICAE (Ge et al., 2024) all use 7B decoder-only models, but these are over 10x larger than our 440M parameter T5 model's decoder and are trained on 10x more tokens. Since we are pretraining DeCAL from scratch, a similar 7B DeCAL is too costly, and comparing across such sizes is not meaningful. We also lack a chunking framework to accommodate the smaller maximum chunk sizes of these models.

---

[1] `https://github.com/google-research/text-to-text-transfer-transformer/blob/main/released_checkpoints.md#t511`

|  | Accuracy |
|---|---|
| Baseline | 70.2 |
| DeCAL 2x | **70.4** |
| DeCAL 4x | 69.2 |
| DeCAL 8x | 68.0 |
| DeCAL 16x | unstable |
| AttnPool 2x | 68.5 |

Table 1: Span corruption pretraining accuracy for different levels of DeCAL compression against vanilla T5 baseline.

We consider it more instructive instead to derive a proxy from the common elements of the retrofitted context compressors for comparison. We find their common elements to be:

1. These all use learnable special tokens appended to the input to aggregate the compressed information.

2. These retrofitted models link the encoder and decoder by passing encoder output as soft tokens into the decoder; in contrast, in the classical encoder-decoder as introduced by the Transformer (Vaswani et al., 2017), and used by T5, each decoder layer cross-attends to the encoder output.

3. Since the encoder is derived from a decoder-only model, it is subject to a causal attention mask.

4. COCOM and ICAE both use autoencoding mixed 1:1 with text continuation for compression training.

5. The encoder is initialized with the decoder weights.

As #5 and #2 are not possible with T5, it being natively encoder-decoder, we use #1, #3, #4 to build the closest proxy we can, which we call *CCProxy*. Specifically, it is constructed similarly to DeCAL, except (1) the latent tokens are appended to the input and a causal mask applied, but without any token pooling, (2) it is initialized with our vanilla T5 baseline, then further trained with a 1:1 autoencoding:text-continuation mix based on C4, for 26M examples. Similarly to DeCAL, we test 2x, 4x, and 8x compression.

We must note that for CCProxy we retained the single learnable vector (with updated position ids) scheme that DeCAL uses. We tried the standard approach of separate learnable vectors per output position with default position ids to better match the retrofitted compressors, but this yielded very poor results. We speculate the reason is that this approach does not scale to the 4k sequence length (with no chunking) we used for evaluation.

Separately, an obvious alternative to prepending a latent sequence to the input is attention pooling the encoder's final output into a shorter sequence. This has lower compute than DeCAL, so it is also worth comparing. We refer to this alternative as *AttnPool* in our results, and only test 2x compression. We train it identically to the DeCAL experiments, and we use mean pool (stride 2) to initialize the query of the attention pool layer.

### 4.2.4 SUMMARIZATION

Table 2 shows ROUGE metrics for the summarization datasets. As with pretraining, we see that DeCAL 2x is always comparable to the baseline, while 4x is only modestly behind. 8x is a notable step lower, but not significantly considering the decoder only cross-attends to 512 tokens versus 4k uncompressed. We speculate DeCAL compression does favorably here because enough of the general semantics of the full input is preserved for the purpose of abstractive summarization.

AttnPool 2x generally comes in close to DeCAL 4x over all the summarization datasets. We see that CCProxy is consistently outperformed by DeCAL at the same compression ratio; like AttnPool 2x, it largely performs similarly to DeCAL at twice the compression ratio.

|  | arXiv | | | PubMed | | | CNN / DailyMail | | | MediaSum | | |
|---|---|---|---|---|---|---|---|---|---|---|---|---|
|  | R-1 | R-2 | R-L | R-1 | R-2 | R-L | R-1 | R-2 | R-L | R-1 | R-2 | R-L |
| Baseline | **45.6** | **18.9** | **41.6** | 48.5 | **23.1** | 45.1 | 42.9 | 20.7 | 37.3 | **35.1** | **18.5** | **32.1** |
| DeCAL 2x | 45.3 | **18.9** | 41.4 | **48.6** | **23.1** | **45.2** | **43.2** | **20.9** | **37.5** | 35.0 | 18.3 | 32.0 |
| DeCAL 4x | 44.9 | 18.6 | 41.0 | 47.9 | 22.5 | 44.5 | 42.2 | 20.1 | 36.7 | 34.1 | 17.6 | 31.2 |
| DeCAL 8x | 44.0 | 17.9 | 40.1 | 47.2 | 21.4 | 43.8 | 41.4 | 19.1 | 35.9 | 32.2 | 16.2 | 29.8 |
| AttnPool 2x | 44.6 | 18.4 | 40.7 | 47.5 | 22.2 | 44.1 | 42.5 | 20.2 | 37.0 | 34.1 | 17.5 | 31.1 |
| CCProxy 2x | 45.0 | 18.4 | 41.0 | 47.6 | 22.3 | 44.2 | 42.5 | 20.2 | 36.7 | 33.6 | 17.2 | 30.5 |
| CCProxy 4x | 44.8 | 18.2 | 40.8 | 47.2 | 21.9 | 43.7 | 41.6 | 19.2 | 35.7 | 33.0 | 16.5 | 29.9 |
| CCProxy 8x | 44.0 | 17.6 | 39.8 | 46.7 | 21.4 | 43.3 | 40.4 | 18.0 | 34.8 | 31.9 | 15.5 | 29.0 |

Table 2: ROUGE metrics on summarization tasks for different levels of compression for both De-CAL and CCProxy against the vanilla T5 baseline and AttnPool 2x.

### 4.2.5 QUESTION-ANSWERING

For the question-answering datasets, presented in Table 3, we see again that DeCAL 2x can fully match the baseline. DeCAL 4x and 8x gradually drop off, but much more so for HotpotQA, likely due to its multi-hop reasoning characteristic. This range of compression ratios gives us a way to trade off compression versus model performance, all the way to 8x.

On these datasets, AttnPool 2x comes close to DeCAL 8x, a worse showing than for summarization. This is a strong indicator that the extra compute cost from adding the latent sequence in DeCAL does translate into superior compressed representations.

These datasets also prove more difficult for CCProxy, with CCProxy 2x matching DeCAL 8x at best. These results establish that the common elements of the retrofitted context compressors can be holding back their performance as compared to a native encoder-decoder design.

|  | TriviaQA | | HotpotQA | |
|---|---|---|---|---|
|  | EM | F1 | EM | F1 |
| Baseline | 76.8 | 79.1 | 65.1 | 79.2 |
| DeCAL 2x | **77.0** | **79.2** | **65.2** | **79.5** |
| DeCAL 4x | 76.0 | 78.3 | 62.9 | 77.0 |
| DeCAL 8x | 75.1 | 77.2 | 60.3 | 74.4 |
| AttnPool 2x | 75.1 | 77.1 | 60.2 | 74.4 |
| CCProxy 2x | 75.1 | 77.3 | 59.2 | 73.6 |
| CCProxy 4x | 74.2 | 76.4 | 56.9 | 71.0 |
| CCProxy 8x | 73.3 | 75.4 | 55.1 | 69.0 |

Table 3: SQuAD metrics on question-answering tasks for different levels of compression for both DeCAL and CCProxy against the vanilla T5 baseline and AttnPool 2x.

### 4.3 ENCODER-ONLY, RETRIEVAL EXPERIMENTS

ColBERT's (Khattab & Zaharia, 2020) late interaction approach offers superior retrieval to single-vector by virtue of allowing each query token to interact with each passage token, but suffers from dramatically greater compute cost (a dot product for each interaction). Pruning techniques have been proposed (Santhanam et al., 2022; Qian et al., 2022) to address the compute cost, but pruning loses information and does not save storage cost. In contrast, DeCAL does not drop any tokens, so information should be lost more gracefully as compression is increased, and by reducing the representation up-front, storage costs are reduced. We do not have a setup for reproducing either

pruning technique for comparison (left for future work), but we include these results to illustrate the versatility of DeCAL and compare against the vanilla T5 baseline.

For these experiments, we took an already pretrained (as described under Setup) DeCAL model at some compression C, discarded the decoder and fine-tuned the encoder in a dual encoder configuration using T5X Retrieval[2]. We utilized Chamfer similarity loss as in ColBERT (Khattab & Zaharia, 2020).

### 4.3.1 DATASETS

We used MS MARCO (Nguyen et al., 2016) as the retrieval fine-tuning dataset (for 12M examples). For evaluation, we used a subset of the BEIR (Thakur et al., 2021) retrieval benchmark: FiQA-2018, TREC-COVID, Touché-2020, NFCorpus, and SciFact, and we report NDCG@10. These (smaller) datasets were selected simply due to resource constraints we had for these experiments.

### 4.3.2 RESULTS

| | FiQA | SciFact | NFCorpus | Touché | COVID |
|---|---|---|---|---|---|
| | NDCG@10 | | | | |
| Baseline | 0.346 | **0.709** | **0.348** | 0.249 | 0.612 |
| DeCAL 2x | **0.359** | 0.703 | **0.349** | **0.299** | 0.685 |
| DeCAL 4x | 0.320 | 0.681 | 0.338 | 0.274 | **0.709** |
| DeCAL 8x | 0.301 | 0.658 | 0.337 | 0.232 | 0.671 |

Table 4: Retrieval results for different levels of DeCAL compression and the vanilla T5 baseline for 5 BEIR datasets.

In Table 4, we see that DeCAL 2x is as good as or better than the baseline, with a 4x computation savings. DeCAL 4x performance drops off only modestly for 16x computation savings. DeCAL 8x drops off further, but at 64x in computation savings and 8x reduction in storage versus the baseline.

Interestingly, we see a wide dispersion of response to compression in the NDCG@10 metric. FiQA is hurt relatively most, but COVID appears to consistently benefit even to 8x compression. We also see that 2x compression is frequently better than no compression; we speculate that compression may be fusing subwords such that the combined information actually improves retrieval. Doing this more intelligently may reap further benefits.

Intuitively, compressing the query too much can hurt multi-vector retrieval since we lose conceptual granularity. We might obtain better results at high compression using an asymmetric dual encoder with low compression on the query, but that is left for future work.

### 4.4 ABLATIONS

In this section we focus on HotpotQA (Yang et al., 2018), since it showed the greatest sensitivity to compression. We also use 2x compression since that represents the upper bound of performance under compression.

There are three main components of DeCAL that we can ablate: (1) language modeling pretraining with a compressive encoder, (2) the latent-based DeCAL compressive structure itself, and (3) the token pooling when forming the initial input latent sequence.

For (1), we can omit the denoising task entirely and first introduce compression during fine-tuning (on top of the vanilla baseline). Alternatively, we can train compression with autoencoding (mixed 1:1 with text continuation to avoid overfitting on reconstruction) in place of denoising, as we did for CCProxy. In Table 5, we see that both of these yield a result that is notably worse than when pretrained with denoising and so can no longer hold up against the baseline.

---

[2]https://github.com/google-research/t5x_retrieval

|  | HotpotQA | |
|---|---|---|
|  | EM | F1 |
| Baseline | 65.1 | 79.2 |
| DeCAL 2x | **65.2** | **79.5** |
| DeCAL 2x (autoencode) | 63.9 | 78.3 |
| DeCAL 2x (fine-tune-only) | 62.6 | 78.0 |
| DeCAL 2x (no pooling) | 61.8 | 75.8 |
| AttnPool 2x | 60.2 | 74.4 |

Table 5: Ablations: SQuAD metrics for HotpotQA comparing (1) the vanilla T5 baseline, (2) De-CAL 2x, (3) autoencoding plus text continuation for DeCAL 2x, (4) applying DeCAL 2x during fine-tune (but not pretrain), (5) omitting token pooling for the initial latent in DeCAL 2x, and (6) AttnPool 2x.

For (2), in place of the DeCAL encoder structure, we can instead apply the commonly-used form of attention pooling by applying it on the final encoder output. This is what we already included in Comparisons as AttnPool 2x. Included here in context with the other ablations, we can see that this is actually the most damaging of them. We conclude that building up the compressed representation over many layers rather than in one is critical for optimal compression.

For (3), When we use only the learnable vector $v$ and skip the initial token pooling, we see that it hurts performance significantly. This indicates that including the representations of the input tokens plays a useful role in guiding the information transfer from the input tokens to the smaller latent sequence.

These ablations give clear support to the value of the multilayer latent-based compressive structure, denoising pretraining, and initial token pooling as important facets of the DeCAL design.

## 5 DISCUSSION

In this section, we tie together some of the above results while doing a deeper comparison of DeCAL with competing methods to highlight its strengths. Table 6 outlines properties of existing context compressors like AutoCompressor (Chevalier et al., 2023), ICAE (Ge et al., 2024), COCOM (Rau et al., 2025). We also include LLoCO (Tan et al., 2024), an AutoCompressor derivative that introduces per-task-type finetuning, and PISCO (Louis et al., 2025), a follow-on to COCOM that invokes teacher distillation in place of pretraining.

Among these, we see that DeCAL has the highest chunk size, 4k tokens, and supports the highest maximum number of output tokens per chunk, 2k, by quite a margin over the next highest at 256 of ICAE. AutoCompressor, with the next highest chunk size, listed as one of its limitations that it saw no improvement past 50 output tokens per chunk. DeCAL is the only compressor that delivers both a larger chunk size and output tokens per chunk. ICAE reported that performance dropped off significantly beyond 4x compression; DeCAL can achieve up to 8x. COCOM and PISCO have the widest compression range, but are limited to QA tasks only and have the smallest chunk size. We have shown DeCAL can be trained as a multi-vector retrieval embedding model, and that it handles summarization tasks well, able to match uncompressed at 2x compression. None of these competing methods were evaluated on summarization tasks except LLoCO (but which did not compare to a finetuned baseline). This indicates that the combination of large chunk size (better contextualization) and output tokens per chunk (more semantic preservation) provided by DeCAL is a valuable factor for supporting summarization with compressed contexts. Hence, DeCAL's native encoder-decoder design customized for compression enables it to occupy a distinct point in the solution space.

It is interesting to note that CCProxy, while lower performing than DeCAL, also shares the same chunk size and max tokens per chunk. If there is a common factor in the retrofitted compressors' designs that limits max tokens per chunk, then CCProxy must lack it. We speculate that the reason is related to our difficulty (covered in Comparisons) when using a separate learnable vector per com-

pressed output token, which fared poorly with the 4k sequence length. The encoder must implicitly learn to route information from input tokens to output tokens via the learnable vectors, and as the number of output tokens rises beyond some point, the space of possible mappings soars, likely rendering the model unable to cope. Whereas CCProxy's use of a single vector and setting of position ids (to the mean of those of the block of corresponding input tokens) constrains the space and makes the problem tractable.

|  | Max chunk size | Max output tokens per chunk | Compression rate |
|---|---|---|---|
| AutoCompressor | 2048 | 50 | 15x-50x |
| LLoCO | 1536 | 50 | 20x-40x |
| ICAE | 512 | 256 | 2x-4x |
| COCOM | 128 | 32 | **4x-128x** |
| PISCO | 128 | 64 | **2x-128x** |
| DeCAL | **4096** | **2048** | 2x-8x |

Table 6: Key properties of different context compression strategies

**Limitations.** Requiring pretraining from scratch makes obtaining large DeCAL encoders very expensive, a large potential limiting factor. However, CEPE (Yen et al., 2024) describes a method to combine a small, separately trained encoder with a larger decoder. CEPE used a 435M parameter encoder (similar in size to ours) and a 7B decoder, so this would be a promising direction for integrating DeCAL's compression capabilities with existing LLMs, and would add flexibility that retrofitted compressors lack. Another limitation is reliance on finetuning per task (also common to most competing methods); ideally, the encoder would be universal across tasks. We see this as a worthy target for future work.

## 6 CONCLUSION

In this paper, we introduced DeCAL, a novel method for tokenwise compression. We demonstrated that the compressed representations can be very usable, matching the uncompressed baseline at 2x compression on all tasks tested. At a substantial 8x compression, average ROUGE-L for DeCAL is only 4.1% lower than baseline for summarization tasks, and average F1 is only 4.3% lower for question-answering tasks. Similarly, average NDCG@10 is only 2.9% lower than baseline on multi-vector retrieval tasks at 8x compression, and demonstrates the versatility of DeCAL over a wide range of downstream tasks.

We identified the key common elements of recent LLM context compressors to construct a proxy for comparison. DeCAL outperformed on both summarization and question-answering tasks, highlighting the advantages of the combination of denoising pretraining, initial token pooling for the latent sequence, and bidirectional attention. We also observed that DeCAL supports a higher chunk size as well as much higher output tokens per chunk compared to recent LLM context compressors.

There are many avenues for further exploration. We found the default T5 (Raffel et al., 2020) span corruption pretrain tasks worked well, but a task tailored for compression may work even better. We have not evaluated the balance of encoder and decoder size or compressing sequences longer than 4k tokens. Work remains to explore techniques to push DeCAL compression to 16x and beyond. We have yet to try Retrieval-Augmented Generation, a common target for context compression, where we can take advantage of pre-computed compressed passages to fill large contexts.

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
