# OpenReview forum: "DeCAL Tokenwise Compression"
_ICLR.cc/2026/Conference — Submitted to ICLR 2026_

### Official Review · Reviewer_juJq · 2025-10-26

**Soundness:** 2
**Presentation:** 1
**Contribution:** 1
**Rating:** 2
**Confidence:** 4

**Summary:**

The paper introduces DeCAL, a tokenwise compression framework that leverages encoder–decoder pretraining with denoising to produce compressed representations without sacrificing downstream performance.DeCAL embeds a learnable latent sequence within the encoder that cross-attends to the evolving input tokens layer by layer, enabling gradual, information-preserving compression. Through systematic experiments across summarization, question answering, and multi-vector retrieval, the authors demonstrate DeCAL's performance.

**Strengths:**

1. The usage of average pooling as the starting point for compressed tokens is interesting.

**Weaknesses:**

1. The work has questionable experiment settings. What are the "baseline" in most of the tables? I assume it's T5, so why don't the authors provide comparison with other token compression works (which can be easily extended to the encoder-decoder setting) such as [1]?
2. L317 - L326, the author mention briefly three token compression methods and claim to use them as baselines, yet no further information is provided to show how they are extended to the paper's experiment setting and only 1 single "baseline" is referenced in the result table. This raises concerns about the clarity on how experiments are conducted.
3. At the end, the goal of most token compression techniques nowadays is to handling long-context tasks, could the authors provide more experiments results under this setting?
4. Why does the work limit the scope of the method under encoder-decoder baselines? Can we extend it to decoder-only architecture by training an additional encoder like [1]? If not, please provide a clear justification.

[1] LLoCO: Learning Long Contexts Offline, Tan et. al., EMNLP 2024.

**Questions:**

Please see Weaknesses.

---

> ### Author Response · Authors · 2025-11-25
> **Response to Reviewer juJq**
>
> Thanks for your feedback and your time!  As we have made substantial updates to our manuscript, we ask to please consider the changes (see the General Response), especially the new Discussion section, and our responses below and let us know what are your remaining areas of concern.
>
> >**W1** The work has questionable experiment settings. What are the "baseline" in most of the tables? I assume it's T5, so why don't the authors provide comparison with other token compression works (which can be easily extended to the encoder-decoder setting) such as [1]?
>
> Apologies for the confusion; we neglected to combine 'uncompressed' with 'baseline' except in one place.  We have made this explicit in all captions that 'Baseline' is always the vanilla T5 model.
>
> We explain in L261-267 why we cannot do direct comparisons.  In short, our model size is much smaller, and we don't have a chunking setup.  We later noticed that the other token compression works do not even report any summarization task metrics, except LLoCO, which unfortunately does not make an apples-to-apples comparison (the baseline is not finetuned while LLoCO is).  We do, however, now compare DeCAL with properties and limitations of the other works in the new Discussion section, that reveal its strengths aside from task metrics.
>
> >**W2** L317 - L326, the author mention briefly three token compression methods and claim to use them as baselines, yet no further information is provided to show how they are extended to the paper's experiment setting and only 1 single "baseline" is referenced in the result table. This raises concerns about the clarity on how experiments are conducted.
>
> We have reworded the CCProxy section to be clearer and more complete, L283-308.  In short, 'baseline' is always vanilla T5, and the three methods mentioned are used to motivate the construction of CCProxy as a best approximation to these methods that DeCAL can be compared against.
>
> > **W3** At the end, the goal of most token compression techniques nowadays is to handling long-context tasks, could the authors provide more experiments results under this setting?
>
> We absolutely agree, and would like to!  However, T5 does not handle long contexts without using LongT5 or similar, which would make the results even less comparable or reproducible.  We need to create a RAG setup to retrieve and encode chunks in parallel, then concatenate to build the long context.  So we plan this as future work.
>
> We do believe our 4k context length for downstream tasks is competitive against ICAE/COCOM/PISCO, which all tested at most 800 tokens of original context.
>
> > **W4** Why does the work limit the scope of the method under encoder-decoder baselines? Can we extend it to decoder-only architecture by training an additional encoder like [1]? If not, please provide a clear justification.
>
> We deliberately used a native encoder-decoder architecture in order to explore what advantages it may have.  So some aspects of the method would not be transferable to what we are calling "retrofitted" models, which train an encoder from the decoder weights.  However, some elements should be fully transferable, such as input token pooling and denoising training, as well as use of a single learnable vector with updated position ids (L173-174).
>
> We also describe how it could be possible to combine a DeCAL encoder with a different, larger decoder, using the method proposed by CEPE [1], see L505-509.  This bypasses the limitation of encoders derived from decoders having to be the same size.
>
> [1]  Long-context language modeling with parallel context encoding, Yen et al.  ACL 2024

---

### Official Review · Reviewer_TQFM · 2025-10-28

**Soundness:** 2
**Presentation:** 2
**Contribution:** 2
**Rating:** 4
**Confidence:** 3

**Summary:**

This paper introduces a new tokenwise compression technique called DeCAL. It follows a T5 architecture, prepending a sequence of “pooling tokens” (summed with average pooling of some input tokens) alongside the input, to extract a compressed version of the input via bidirectional self-attention. This compressed input sequence is fed to a decoder. It follows pre-training on a span denoising task, and finetunes on several QA, summarisation and retrieval tasks. It shows good downstream performance on those tasks with compression, with a performance with 2x compression matching the performance of the “uncompressed” baseline.

**Strengths:**

- The paper summarizes well the different related works on compression of token sequences.
- The paper is easy to follow.
- The paper present a significant number of experiments: from pre-training with different compression ratios, experiments with others methods like AttnPool or the custom CCProxy approaches, to a number of finetuning experiments on several tasks including Document-based QA, Summarization and Retrieval.
- The paper presents a few ablations on modelling choices in section 4.4
- The paper shows good results in comparison with the baseline, while using high compression ratios, on many tasks

**Weaknesses:**

The paper method lacks of novelty:
 - Extracting compressed representation with self-attention has been widely used in other works (cited and presented by the authors) like COCOM with CTX tokens as well as ICAE with memory-tokens.
 - Span denoising tasks with encoder-decoder models has also been widely studied

Novelty lies in the combination of these two, but a reader would expect a more thorough analysis of why this method could be superior to previous work, what component is critical, what percentage of masking is required, are there other denoising tasks that could work better.

While the authors have included baselines like CCProxy with the same training setup, the paper would greatly benefit from external baselines.

The paper would also benefit from RAG setup experiments with the presented method for bigger potential impact.

A few explanations in text do not seem to match figures/table caption: e.g. Figure 1. "init. using pooled input tokens" while in text the pooled representations are added to learnable vector. Table 6. DeCAL 2x (auto-encode) presented as "denoising replaced by auto-encoding" is presented in the text as "auto-encoding (mixed with text continuation)"

**Questions:**

Can you provide more details on the CCProxy setting ? ("start from our baseline model" + "decoder-only model" --> Which part of the model do you use?). The training data setting seems different also (128M vs. 26M C4 examples)

Why do you repeat the same learnable vector? Did you try having several learnable vectors? How did you initialize this vector?

Did you evaluate the representations obtained from pre-training (keeping the encoder frozen), while fine-tuning only the decoder on downstream tasks?

Did you try other span masking strategies / ratios?

---

> ### Author Response · Authors · 2025-11-25
> **Response to Reviewer TQFM**
>
> Thanks for your feedback and your time!  As we have made substantial updates to our manuscript, we ask to please consider the changes (see the General Response), especially the new Discussion section, and our responses below and let us know what are your remaining areas of concern.
>
> >**W1** The paper method lacks of novelty:
> Extracting compressed representation with self-attention has been widely used in other works (cited and presented by the authors) like COCOM with CTX tokens as well as ICAE with memory-tokens.
> Span denoising tasks with encoder-decoder models has also been widely studied
> Novelty lies in the combination of these two, but a reader would expect a more thorough analysis of why this method could be superior to previous work, what component is critical, what percentage of masking is required, are there other denoising tasks that could work better.
>
> To be clear, our exposition about the latent sequence attention is to contrast it with compression techniques like Funnel Transformer.  Our contributions list does not claim this part is itself novel.
>
> We list more thoroughly in the General Response the novel aspects of our method, which is much more than denoising.  Hopefully that addresses any concerns about novelty.  We are happy to clarify further.
>
> Based on your feedback, we did add an explanation as to why denoising would be expected to work better than autoencoding (L151-157).  Hopefully that adds some insight.  Since denoising is only one part of our overall method, we didn't delve extremely deeply.
>
> >**W2** While the authors have included baselines like CCProxy with the same training setup, the paper would greatly benefit from external baselines.
>
> We agree that having direct comparison metrics of competing methods would strengthen our paper.  We have expanded our explanation on L261-267 why we do not (in short, our model size is much smaller, and we don't have a chunking setup).
>
> In lieu of task metrics we are able to compare properties and limitations of other methods and highlight advantages of DeCAL (like its much larger output token capability than all other methods), which we have added in the new Discussion section, L463-502.
>
> >**W3** The paper would also benefit from RAG setup experiments with the presented method for bigger potential impact.
>
> We absolutely agree!  However, we don't have a RAG setup, so must defer that to future work.
>
> >**W4** A few explanations in text do not seem to match figures/table caption: e.g. Figure 1. "init. using pooled input tokens" while in text the pooled representations are added to learnable vector. Table 6. DeCAL 2x (auto-encode) presented as "denoising replaced by auto-encoding" is presented in the text as "auto-encoding (mixed with text continuation)"
>
> Thanks for the corrections!  We have removed the inconsistencies in our revision.
>
> >**Q1** Can you provide more details on the CCProxy setting ? ("start from our baseline model" + "decoder-only model" --> Which part of the model do you use?). The training data setting seems different also (128M vs. 26M C4 examples)
>
> Apologies for the lack of clarity.  We have revamped the CCProxy description to address this (L283-308).  128M is for training the baseline model, which is the starting point of autoencoding + text continuation training (26M) to obtain CCProxy.  It is encoder-decoder throughout (being on T5).
>
> >**Q2** Why do you repeat the same learnable vector? Did you try having several learnable vectors? How did you initialize this vector?
>
> We repeat it because it allows generalization to more output tokens after pretraining (more details L210-214).  For DeCAL, we never tried separate vectors.  We did try this on CCProxy to address a question by Reviewer ka4m, and found it worked surprisingly poorly.  We discuss this interesting result on L304-308 and L483-491.
> The vector was initialized as a normal distribution centered at 0 with std. deviation of 1.
>
> >**Q3** Did you evaluate the representations obtained from pre-training (keeping the encoder frozen), while fine-tuning only the decoder on downstream tasks?
>
> No, we never tried that, but that is definitely a worthy thing to try in the future!
>
> >**Q4** Did you try other span masking strategies / ratios?
>
> We only tried 10% and 20% in addition to 15%, but found no obvious effect on pretraining.  That was early on, though, and before we had all the downstream tasks set up.

---

> > ### Comment · Reviewer_TQFM · 2025-11-27
> >
> > I appreciate the response of the authors (including the general response).
> >
> > However, I still feel like novelty is too limited for an ICLR paper. Moreover, the lack of external baselines makes the results difficult to compare with previous work methods.
> >
> > Nevertheless, I still recognise the strengths highlighted in my original review.
> >
> > Therefore, I maintain an overall score of 4.

---

### Official Review · Reviewer_ka4m · 2025-11-01

**Soundness:** 3
**Presentation:** 3
**Contribution:** 2
**Rating:** 4
**Confidence:** 4

**Summary:**

This paper propose a new method to learn compressed representation of text. It considers a setting with an encoder-decoder model, with the goal of reducing the number of hidden representations from the encoder that are fed to the decoder. More precisely, the paper proposes a modification of the T5 encoder-decoder architecture. For a sequence of N tokens, the proposed method, called DeCAL, will compress the sequence to N/C continuous representations, where C is the compression factor. These compressed representations are obtained by adding N/C embeddings to the input of the encoder. These embeddings are obtained by applying mean-pooling to the token embeddings of the input sequence. A learnable embedding is also added to each mean pooled representation. Then, this augmented sequence is processed normally by the T5 encoder. The only modification are the position bias: for the additional tokens, the integer average of the position of the original tokens are used. Then, the output corresponding to the added tokens are used as input of the cross attention of the decoder, which is unchanged. The model is then pre-trained with the same denoising task as in T5, and fine-tuned on downstream tasks such as QA or summarization. The method is then evaluated on summarization, question answering and retrieval tasks. An ablation study is performed to study the impact of pre-training, of the task used to train the model (T5 denoising vs. reconstruction) and different ways to obtain the compressed representations, showing that the different components are important.

**Strengths:**

Overall, I believe that this paper is borderline regarding acceptance to ICLR.

On the positive side, I think that the paper is well written and easy to follow. The proposed method is simple and sound, and the practical implementation is solid. I enjoyed the ablation experiments, and believe that its conclusion could impact the future development of learning compressed text representation. More precisely, the paper show that:
- training an encoder from scratch leads to better results than fine-tuning a model pre-trained without compression ;
- initializing the compressed representations from pooled embeddings of the text tokens leads to significantly better results than learned parameters only, such as memory tokens that were previously considered (here, it would be interesting to compare to a version with different parameters for each position of the compressed representation) ;
- the experiments show that using this way to obtain compressed representations leads to significantly better results than doing a attention pooling at the end of the encoder, hence showing that using more "compute" to learn the compressed representation leads to better results. I am wondering how this results is linked to the fact that the model is pre-trained in this way, and if the result would still be true in the context of fine-tuning only.

**Weaknesses:**

But on the other hand, I also believe that the paper suffers from some limitations.

First, as currently proposed in the paper, the DeCAL method is strictly more computationally expensive than the T5 baseline, or other methods used as comparison in the paper. This limits the usability in practice of DeCAL, and I think that it would make the paper stronger to discuss (and potentially explore) how this could be useful in real world settings.

A second limitation is that, I believe that a different model was fine-tuned for each downstream task. I think it would have been more interesting to explore if a single model could learn to compress text representations for multiple downstream tasks. I think that this is important, as the information needed for different downstream tasks could be quite different. Thus, learning a single model that compresses well for many tasks is potentially more challenging than learning to compress individually for each task. I believe that most previous work learn a single model for multiple tasks.

Finally, I feel that the comparisons to previous work is a bit weak. There has been multiple papers exploring how to obtain compressed representations, and this paper only compared to a "Proxy" method, where the authors chose the key elements of previous work. To me, this comparison feels more like an ablation than a comparison to existing methods. In particular, the setting considered in the paper is quite different from what is considered by others (see previous point), potentially making the comparison unfair.

**Questions:**

For the CCProxy baseline, did you use a single learnable vector shared for all the positions?

Did you try the DeCAL method in a multitask setting?

Additional references:

- J. Mu, X. L. Li, N. Goodman. Learning to compress prompts with gist tokens, 2024
- J. Tang, Z. Zhang, et al. GMSA: Enhancing context compression via group merging and layer semantic alignment, 2025
- S. Eyuboglu, R. Ehrlich, et al. Cartridges: Lightweight and general-purpose long context representations via self-study, 2025
- M. Louis, H. Dejean, S. Clinchant. Pisco: Pretty simple compression for retrieval-augmented generation, 202

---

> ### Author Response · Authors · 2025-11-26
> **Response to Reviewer ka4m  Part 1**
>
> Thanks for your feedback and your time!  A couple of your comments led to further analysis and material updates, improving our work overall.
>
> As we have made substantial updates to our manuscript, we ask to please consider the changes (see the General Response), especially the new Discussion section, and our responses below and let us know what are your remaining areas of concern.
>
> >**W1** First, as currently proposed in the paper, the DeCAL method is strictly more computationally expensive than the T5 baseline, or other methods used as comparison in the paper. This limits the usability in practice of DeCAL, and I think that it would make the paper stronger to discuss (and potentially explore) how this could be useful in real world settings.
>
> Actually, all the other similar compression methods (Autocompressor, ICAE, COCOM, and derivatives) add extra tokens and thus have the same increase in encoder computation.  We guess they just gloss over it (since it is minor at high compression ratios).  We bring it more to light since we also compare against simple attention pooling as a sequence reduction mechanism.
>
> Also, the extra cost can be amortized if the compressed representations can be stored and reused.  This is precisely why we chose to test DeCAL on multi-vector retrieval, a practical real world setting where the embeddings are naturally stored.  Here, DeCAL can offer significant storage space reduction and retrieval-time computation savings with minimal loss in retrieval metrics, making the extra compute worthwhile.
>
> >**W2** A second limitation is that, I believe that a different model was fine-tuned for each downstream task. I think it would have been more interesting to explore if a single model could learn to compress text representations for multiple downstream tasks. I think that this is important, as the information needed for different downstream tasks could be quite different. Thus, learning a single model that compresses well for many tasks is potentially more challenging than learning to compress individually for each task. I believe that most previous work learn a single model for multiple tasks.
>
> You are correct in that we did finetune for each task.  We were following common procedure in similar T5 papers and didn't otherwise think about it, so we are glad you brought it to our attention!  We wholeheartedly agree that a single model that served many tasks would be very useful and worthwhile, and we will make this a target for future exploration.
>
> We did look at the prior works to see what they did, and the answer is they also mostly finetune, but usually not at a per-task granularity, but by type of task, finetuning on groups of similar datasets (like question-answering).
> Specifically,
> 1. Autocompressor does not finetune, so it is generic.
> 2. LLoCO, however, adds finetuning to Autocompressor showing significant metric gains.  They state: "For each dataset we evaluate, we perform instruction finetuning using the question-answering pairs from the training set"
> 3. ICAE finetunes on its multitask PwC dataset (limited to 800 context tokens in length), but keeps the decoder frozen.  It doesn't evaluate on any particular benchmark datasets, however, so we can't identify its strengths and weaknesses by task type.
> 4. COCOM finetunes on a mix of question-answering datasets, filtering out contexts > 128 tokens.
> 5. PISCO trains on teacher labels on top-5  retrieved Wikipedia passages (max 128 tokens each) plus question as context.
>
> So in comparison to these, we don't think our per-task finetuning is that much of a weakness.  We are similar to LLoCO, we cover more task types than COCOM/PISCO, and we report more task metrics than ICAE.
>
> Actually, after examining all these works closely, we realized that only LLoCO even evaluates on any summarization tasks (though it compared to a non-finetuned baseline).  This void of summarization evaluations is most likely due to the limited chunk sizes of these context compressors.  These limitations probably need to be tackled first before we will have properly multitask compressors that handle longer contexts.

---

> ### Author Response · Authors · 2025-11-26
> **Response to Reviewer ka4m Part 2**
>
> >**W3a** Finally, I feel that the comparisons to previous work is a bit weak. There has been multiple papers exploring how to obtain compressed representations, and this paper only compared to a "Proxy" method, where the authors chose the key elements of previous work. To me, this comparison feels more like an ablation than a comparison to existing methods.
>
> We agree that having direct comparison metrics of competing methods would strengthen our paper.  We have expanded our explanation on L261-267 why we do not (in short, our model size is much smaller, and we don't have a chunking setup).
>
> In lieu of task metrics we are able to compare properties and limitations of other methods and highlight advantages of DeCAL (like its much larger output token capability than all other methods), which we have added in the new Discussion section, L463-502.
>
> >**W3b** In particular, the setting considered in the paper is quite different from what is considered by others (see previous point), potentially making the comparison unfair.
>
> If by 'setting' you mean per-task finetuning, as discussed earlier, we don't think it is materially different.  If not, could you elaborate what you mean?  Thank you.
>
> >**Q1** For the CCProxy baseline, did you use a single learnable vector shared for all the positions?
>
> Good question!  It turns out we did--we had overlooked this aspect, but the intention was to approximate the similar context compressors as much as possible.  After your question, we made the change to use separate vectors for the output tokens, but the resulting metrics were abysmal.  So CCProxy was left as is, and we added details in L304-308. We speculate how this may be connected to the limitations we observed on max output tokens of the retrofitted context compressors in L483-491.  Hopefully this finding might help those compressors overcome that limitation.
>
> >**Q2** Did you try the DeCAL method in a multitask setting?
>
> No, we did not. That is certainly worthwhile to try, but we probably should use larger models for that, so we'll look into this in the future once we scale up DeCAL.

---

### Official Review · Reviewer_7Zks · 2025-11-02

**Soundness:** 2
**Presentation:** 3
**Contribution:** 1
**Rating:** 2
**Confidence:** 5

**Summary:**

Authors propose a compression method based on encode-decoder architecture and showed the evaluation of the method on several benchmark datasets. They have performed ablation on compression ratio and also compared with other methods.

**Strengths:**

- performed experiments on a several benchmark tasks
- shown variations on different compression ratios
- comapred with other compression methods
- paper is well written and easy to follow

**Weaknesses:**

- The comparson is limited to CCProxy and AttnPool 2x. The justification for CCProxy hyper-parameters are not good enough (variations in the models should be used, ablation needed). The existing methods like NUGGET show much higher performances. Also the experiments are shown on two datasets only. Ablaton needed.
- What if the T5 as a basemodel is changed? is this method generalizable to other architectures? Ablaton neeed.
- for summarization/ tasks, more useful metrics should be used

**Questions:**

- What creates the unstability in training?
- What variations are possible in token selection?

---

> ### Author Response · Authors · 2025-11-25
> **Response to Reviewer 7Zks**
>
> Thanks for your feedback and your time!  As we have made substantial updates to our manuscript, we ask to please consider the changes (see the General Response), especially the new Discussion section, and our responses below and let us know what are your remaining areas of concern.
>
> >**W1a** The comparson is limited to CCProxy and AttnPool 2x. The justification for CCProxy hyper-parameters are not good enough (variations in the models should be used, ablation needed).
>
> We tested matching compression ratios for both DeCAL and CCProxy.  Otherwise, could you please be more specific about  what variations and hyper-parameters you mean?  Thank you!
>
> >**W1b** The existing methods like NUGGET show much higher performances.
>
> Could you please elaborate in which ways NUGGET has higher performance?  We did not see any generative downstream tasks being evaluated for NUGGET, just perplexity on language modeling.  For that, NUGGET also did not use the same history length for direct comparison to the uncompressed baseline--NUGGET had the most recent 128 tokens uncompressed + the 128 prior tokens compressed, compared to just the most recent 128 tokens uncompressed as baseline, ie. effectively 256 vs 128.  In contrast, DeCAL compressed the entire context in all experiments (including pretraining), and the baseline and all DeCAL variants had identical input contexts.  Lastly, we want to point out that NUGGET is limited to a chunk size of 128 tokens vs 4096 for DeCAL.
>
> >**W1c** Also the experiments are shown on two datasets only. Ablaton needed.
>
> We have now included CCProxy results for all the encoder-decoder datasets.  Please see the updated tables.
>
> >**W2** What if the T5 as a basemodel is changed? is this method generalizable to other architectures? Ablaton neeed.
>
> The current method is tied to use of a native encoder-decoder architecture, but BART could be used instead.  Some of the DeCAL design elements should be fully transferable, such as input token pooling and denoising training, as well as use of a single learnable vector with updated position ids (L173-174).
>
> We also describe how it could be possible to combine a DeCAL encoder with a different, larger decoder, using the method proposed by CEPE [1], see L505-509.  This bypasses the limitation of encoders derived from decoders having to be the same size.
>
> >**W3** for summarization/ tasks, more useful metrics should be used
>
> If you’re referring to using an LLM to judge summaries, we acknowledge the value, but we do not have such a setup in our eval framework currently.  We do believe that the metrics we used suffice to allow comparisons internal to our experiments given they are all variants built off T5 Large trained on C4 (not disparate models) and compression rates are modest.
>
> We also note that another reviewer referenced LLoCO (EMNLP 2024), which upon inspection uses LongBench, and in LongBench’s Table 1, it shows the metrics used are Rouge-L for summarization, which is the same as ours.  This is the only competing method that even evaluated summarization.
>
> >**Q1** What creates the unstability in training?
>
> Good question!  We think that compressing beyond a certain level interferes with the denoising task.  Most likely, multiple spans to reconstruct are falling into the same compressed block (i.e. same output token).  We are looking into ways to address this in future work.
>
> >**Q2** What variations are possible in token selection?
>
> If you mean: "Can the input pooling be done other than with contiguous blocks of the same size?", then, yes, we could in principle use any mapping, such as mapping output tokens to sentences. Exploring this is future work.
>
> [1]  Long-context language modeling with parallel context encoding, Yen et al.  ACL 2024

---

### Author Response · Authors · 2025-11-25
**General Response**

We thank all reviewers for their comments!

Based on the collective feedback, we have made substantial improvements to the manuscript, with added and modified sections highlighted in blue.

* **Clarity**
   * Introduced terminology of 'retrofitted' and 'native' to distinguish two types of encoder-decoder architectures to better differentiate DeCAL.
   * Clarified the meaning of baseline throughout, including in captions.  (It is always the uncompressed vanilla T5 model.)
   * Expanded reason for proxy instead of direct comparisons, L261-267
   * Expanded the details of CCProxy, L283-308
   * Clarified that all experiments have identical input context, L247-249
* **Exposition**
    * Added an explanation why denoising should be better than autoencoding for compression, L151-157
    * Added an explanation for why DeCAL uses a single learnable vector, L211-214
    * Added our findings when testing separate learnable vectors per token for CCProxy, L304-308
    * Added discussion section to compare against properties of competing methods, L463-502
    * Added discussion of limitations of DeCAL, L504-511
* **Experiments**
    * In the interim since submittal, we obtained results for CCProxy for **all** our encoder-decoder datasets instead of just two, so these are now included.

* * *
\
Separately, we recognize the common concerns from reviewers about **novelty/contribution**. We realize that we did not do a good job conveying this well and have worked to improve this in the new revision.  In particular, we added a Discussion section that goes more in-depth covering aspects of DeCAL compared to prior works and their limitations.

For completeness here, among all of AutoCompressor, LLoCO, ICAE, COCOM, PISCO, (the methods that compress via extra tokens), we have shown that DeCAL is the **sole method** that:

1. Pretrains with compression from scratch
2. Utilizes a denoising task
3. Token-pools the initial input
4. Uses a native encoder-decoder architecture (bidirectional encoder + encoder-decoder cross-attention)
5. Was applied to multi-vector retrieval tasks
6. Was evaluated on summarization tasks
7. Has the highest demonstrated chunk size (improving contextualization) and, especially, max output tokens per chunk (semantic preservation).

\#1-\#3 we have demonstrated contribute to improved performance via ablations, \#4 is partly simulated by CCProxy (effect of bidirectional attention).

For \#6, note that LLoCO did evaluate on summarization tasks, but was finetuned on summarization datasets, while their baseline was not, so we don't count it as properly evaluated.

Regarding \#7, AutoCompressor has the next largest chunk size and cites no improvement from increasing its limited 50 tokens per chunk, lower than both ICAE and COCOM, as though chunk size and tokens per chunk are inversely related.  Our attempt to apply separate learnable tokens to CCProxy (to address a question by Reviewer ka4m) was a failure, but may hint at the underlying reasons for this common tokens-per-chunk limitation (we provide our insight in the Discussion section L483-491).

We hope the above makes clear the several unique aspects of DeCAL and its contribution to the space of context compression, hence value to the research community.

---

### Meta-Review · Area_Chair_Wv6s · 2026-01-05

**Summary:**

**Summary of the paper**

This paper proposes DeCAL, a token-wise compression method built on a modified T5 encoder–decoder architecture. DeCAL compresses an input sequence of length (N) into (N/C) continuous representations by prepending pooled latent tokens to the encoder input and using their final representations as the compressed output. The model is pretrained from scratch using T5-style span denoising and then fine-tuned on downstream tasks including summarization, question answering, and multi-vector retrieval. Experimental results show that DeCAL at moderate compression ratios (e.g., 2×) can match or closely approach the uncompressed T5 baseline, with gradual degradation at higher compression. Ablation studies investigate the roles of denoising pretraining, token pooling, and architectural choices.

**Summary of the reviewers' concerns**

Reviewer ka4m provided a detailed and balanced assessment, describing the paper as borderline but identifying several substantive limitations. While acknowledging that the method is sound, well written, and supported by careful ablations, the reviewer emphasized that DeCAL is strictly more computationally expensive than the T5 baseline and other comparison methods, limiting its practical utility. The reviewer also raised concerns that a separate model is fine-tuned per downstream task, whereas much prior work aims to learn a single compressed representation usable across multiple tasks. Finally, Reviewer 1 found the comparisons to prior compression methods weak, noting that the CCProxy baseline feels closer to an ablation than a true comparison to existing approaches, especially given differences in experimental settings.

Reviewer TQFM expressed stronger concerns about novelty. This reviewer noted that both key ingredients of DeCAL, self-attention–based token compression and span-denoising pretraining, are well established in prior work, and that the contribution appears to lie mainly in their combination. Reviewer 2 emphasized the lack of external baselines beyond CCProxy and AttnPool, questioned clarity and consistency in some figures and tables, and suggested that evaluation in retrieval-augmented generation (RAG) settings would substantially strengthen the paper.

Reviewer juJq was more critical overall, rating the contribution and presentation as poor. This reviewer questioned the experimental clarity, particularly what constitutes the “baseline” in the tables and how other token compression methods are incorporated or approximated. They also expressed concern that the paper does not evaluate DeCAL on long-context settings, which are a central motivation for token compression, and questioned the restriction to encoder–decoder architectures.

**Reviewer Concerns:**

The rebuttal partially addresses several clarity and positioning issues raised by Reviewers ka4m and TQFM. The authors explain that the increased encoder compute is shared by other latent-token compression methods and argue that DeCAL’s cost can be amortized in settings such as multi-vector retrieval where compressed representations are stored. They also provide a detailed discussion situating DeCAL relative to prior work (e.g., AutoCompressor, ICAE, COCOM, PISCO, LLoCO) and clarify why direct metric-level comparisons are difficult due to differences in model size and chunking setups. Several textual inconsistencies and ambiguities (e.g., CCProxy details, figure/table descriptions) are also corrected.

However, key concerns remain unresolved. Most notably, the rebuttal does not fully overcome concerns about incremental novelty raised by Reviewer TQFM: while the authors clarify that certain components are not individually claimed as novel, the paper still lacks a compelling, mechanistic explanation of why DeCAL’s design should outperform prior token compression methods beyond empirical results. The limitation of per-task fine-tuning, highlighted by Reviewer ka4m, is acknowledged but deferred to future work.

**Reviewer Scores:**

Reviewers ka4m and TQFM are likely to keep their boarderline scores.  Reviewer juJq may increase the score, but still less likley.

---

### Decision · Program_Chairs · 2026-01-26

Reject